# OpenReview forum: "Inverse Scaling Can Become U-Shaped"
_EMNLP/2023/Conference — EMNLP 2023 Main_

### Official Review · Reviewer_Fce9 · 2023-07-26

**Soundness:** 3

**Excitement:**

4: Strong: This paper deepens the understanding of some phenomenon or lowers the barriers to an existing research direction.

**Paper Topic And Main Contributions:**

This paper studies the inverse scaling behavior of language models and finds that the declining performance increases again as the model continues to scale, (i.e., U-shaped scaling). The author hypothesizes that the cause of this phenomenon is the existence of distractors. Accordingly, the author proposes to mitigate the inverse scaling behavior by providing demonstrations or chain-of-thought. In all evaluated tasks, this simple approach turns the reverse scaling performances into a U-shape.

**Questions For The Authors:**

1. As the potential explanation implies, when models are small, larger models are more likely to be confused by the distractors. Could you provide further discussion to explain this?
2. What could be potential solutions to left-shift the bottom of U?


**Reasons To Accept:**

1. The finding of U-shape performance is highly worth noticing.
2. And the authors provide a reasonable explanation for inverse scaling, which they further support with a mitigation strategy.
3. The experiments are extensive and convincing, involving diverse inverse-scaling tasks and different models.


**Reasons To Reject:**

1. The prompting method highly relies on manual construction.
2. Lack of direct support for the proposed explanation.

**Reproducibility:**

3: Could reproduce the results with some difficulty. The settings of parameters are underspecified or subjectively determined; the training/evaluation data are not widely available.

**Reviewer Confidence:**

4: Quite sure. I tried to check the important points carefully. It's unlikely, though conceivable, that I missed something that should affect my ratings.

---

> ### Author Rebuttal · Authors · 2023-08-28
>
> Thanks so much for your review!
>
> > The prompting method highly relies on manual construction.
>
> This is true, and this is definitely a limitation of our proposed mitigation strategies. However, as we discuss in our response to Reviewer 95LF, the manual construction was not high-effort, and the method only requires a construction of just a single demonstration. We believe this is an interesting area for future improvement—zero-shot CoT, as Reviewer 95LF suggested, could be a potentially effective strategy, and pseudo-demonstrations (Lyu et al. 2023: https://arxiv.org/abs/2212.09865) could also be explored in the future.
>
> > Lack of direct support for the proposed explanation.
>
> We agree that finding empirical support for the proposed explanation is important and we hope to explore this in more detail in future work. Qualitative analysis of the errors, which we plan to include in the next revision, could potentially provide insights.
>
> > As the potential explanation implies, when models are small, larger models are more likely to be confused by the distractors. Could you provide further discussion to explain this?
>
> We are not sure we fully understood “when models are small, larger models are more likely to be confused by the distractors”, but assuming this is asking why larger models are more likely to be confused by distractors, our rationale is that sensitivity to the distractor task requires a degree of understanding that smaller models may lack. And because understanding and correctly solving the distractor task entails that the model answers will be incorrect in light of the true task (definition of a distractor task), larger models will perform more poorly than the smaller models.
>
> > What could be potential solutions to left-shift the bottom of U?
>
> As can be seen from Figure 3, CoT prompting does left-shift the bottom of the U for some tasks (e.g., Into the Unknown: 62B -> 1B, Negation QA: 62B -> 8B). While CoT isn’t a universal solution, it would be a good starting point to build future solutions upon—e.g., meta-reasoning approach proposed in Yoran et al. (https://arxiv.org/abs/2304.13007).

---

### Official Review · Reviewer_95LF · 2023-07-27

**Typos Grammar Style And Presentation Improvements:** The writing is great.
**Soundness:** 4

**Excitement:**

4: Strong: This paper deepens the understanding of some phenomenon or lowers the barriers to an existing research direction.

**Missing References:**

There is limited discussion of related works.

**Paper Topic And Main Contributions:**

The authors identify that many "inverse scaling" tasks actually exhibit U-shaped performance curves when we consider previously unexplored large model scales, i.e., PaLM 540B. The authors provide a plausible explanation for this, suggesting that medium-sized models are confused by distractor tasks, such as repetitive patterns, completing famous quotes, recent instructions, etc. Based on this intuition, the authors propose two ways to mitigate inverse scaling behavior by clarifying the distractor/true task, namely one-shot prompting and one-shot chain-of-thought prompting, which are shown to be quite effective.

**Questions For The Authors:**

- A. It would be interesting to see the effect of instruction tuning on the models' ability to follow the *true* task. Were there any experiments on this?
- B. It would be interesting to see if zero-shot CoT [Kojima 2022] can help the model overcome the distractor tasks through step-by-step reasoning. This may help isolate the effect of (1) task guidance/position and (2) step-by-step reasoning / introspection elicited by few-shot CoT. Zero-shot CoT has the potential to be a generalizable mitigation strategy discussed in the limitations. Were there any experiments on this?
- C. Can you explain the prompt engineering efforts that went into constructing the CoT prompts, if any? The level of effort needed to achieve the performance gains shown in the paper is somewhat important to understand the applicability of the mitigation strategy. Also, details/findings from the prompt engineering process may provide helpful insights for readers. This is not discussed in Appendix C.

**Reasons To Accept:**

- The paper identifies an interesting phenomena regarding the performance scaling and quirks of LLMs. The findings contradict the results of the inverse scaling challenge which has garnered much attention.
- The paper brings attention to a plausible explanation for the U-shaped scaling curves and suggests effective ways to mitigate the unwanted performance degradations, i.e. inverse scaling.
- The presentation is very clear.

**Reasons To Reject:**

- A. The results are not reproducible due to the use of PaLM, a proprietary model. Though, arguably, the paper makes an important contribution by bringing forth these findings that would otherwise not be discoverable with current public models. GPT-4 is a possible avenue, but limited details on the model would hinder rigorous study. Side note: it would be interesting to see results on GPT-4.
- B. Qualitative analysis on the success/failure cases is not considered. This would help support the proposed hypothesis that there are *true* tasks and *distractor* tasks, and possibly give more insights into how larger models or proposed techniques overcome inverse scaling. Such analysis would be very feasible considering the number of tasks (11).

**Reproducibility:**

2: Would be hard pressed to reproduce the results. The contribution depends on data that are simply not available outside the author's institution or consortium; not enough details are provided.

**Reviewer Confidence:**

4: Quite sure. I tried to check the important points carefully. It's unlikely, though conceivable, that I missed something that should affect my ratings.

---

> ### Author Rebuttal · Authors · 2023-08-28
>
> Thanks so much for your review!
>
> > The results are not reproducible due to the use of PaLM, a proprietary model.
>
> PaLM, while not completely open-source, is now accessible via the PaLM/Google Cloud API (https://developers.generativeai.google/).
>
> > Side note: it would be interesting to see results on GPT-4.
>
> We agree, and there are some partial results in the GPT-4 technical report, where GPT-4 seems to do quite well. However, it is unclear what the significance of the results from GPT-4 is in the context of scaling, since we do not know the model size or compute used to train GPT-4.
>
> > Qualitative analysis on the success/failure cases is not considered. This would help support the proposed hypothesis that there are true tasks and distractor tasks, and possibly give more insights into how larger models or proposed techniques overcome inverse scaling.
>
> This is a great idea and we will try to add qualitative analysis to the camera-ready, if the paper were to be accepted.
>
> > It would be interesting to see the effect of instruction tuning on the models' ability to follow the true task. Were there any experiments on this?
>
> We do not currently have experimental results on the effects of instruction tuning, but we could evaluate the Flan-PaLM series to measure this effect.
>
> > It would be interesting to see if zero-shot CoT [Kojima 2022] can help the model overcome the distractor tasks through step-by-step reasoning. This may help isolate the effect of (1) task guidance/position and (2) step-by-step reasoning / introspection elicited by few-shot CoT. Zero-shot CoT has the potential to be a generalizable mitigation strategy discussed in the limitations. Were there any experiments on this?
>
> This is a great idea and we’d love to include results testing zero-shot CoT to Table 2. Thanks for the suggestion!
>
> > Can you explain the prompt engineering efforts that went into constructing the CoT prompts, if any? The level of effort needed to achieve the performance gains shown in the paper is somewhat important to understand the applicability of the mitigation strategy. Also, details/findings from the prompt engineering process may provide helpful insights for readers. This is not discussed in Appendix C.
>
> We did not extensively prompt engineer the CoT prompts—for most prompts, one of the authors came up with one step-by-step formulation of the problem and that was sufficient. We can add this discussion to Appendix C.
>
> > There is limited discussion of related works.
>
> We can add more discussion of related work using the additional page allowed in the camera-ready. If there is important related work that you’d like to suggest including, please feel free to share the references!

---

### Official Review · Reviewer_bS6X · 2023-07-30

**Soundness:** 4

**Excitement:**

4: Strong: This paper deepens the understanding of some phenomenon or lowers the barriers to an existing research direction.

**Paper Topic And Main Contributions:**

This paper delves into the phenomenon of inverse scaling in language models and brings attention to how larger pre-trained models can counter the trends observed by McKenzie et al. The presence of U-shaped curves suggests that relying on inverse scaling laws to predict a model's performance during training may not be accurate, as the performance could decrease, stabilize, or even improve.

Moreover, the study explores different prompting strategies, such as 1-shot in-context learning and chain-of-thought reasoning, aiming to tackle challenging tasks effectively with sufficiently large language models. The experimental results provide supporting evidence that the effect of inverse scaling can be mitigated when models are guided through demonstrations, enabling them to steer clear of "distractor tasks" that are concealed within each task in the Inverse Scaling Prize benchmark.

**Questions For The Authors:**

Question A: Regarding the prompts shown in Figure 5, I am curious about the potential impact of the target label order on the models' performance. Zhao et al. [1] have discussed tendencies like a recency bias (models favoring the last target label) and a "common token" bias (frequently occurring answers from the pre-training data). Have you verified whether the labels are evenly distributed in the prompts for each task, ensuring that the correct answer is not disproportionately positioned as the first or second option?

Zhao et al. "Calibrate Before Use: Improving Few-shot Performance of Language Models." In Proceedings of the 38th International Conference on Machine Learning (ICML'21). https://proceedings.mlr.press/v139/zhao21c.html



**Reasons To Accept:**

This study challenges the inverse scaling law, which indicated that an increase in pre-training compute leads to a decrease in performance. The authors demonstrate that certain models defy this scaling law, rendering it unsuitable for reliable extrapolation as an indicator of future model performance. Furthermore, the authors illustrate that adequately large language models are able to identify the "true task" from a single in-context example, while also presenting possibilities for future research on imitation strategies that eliminate the need for task-specific tuning or example selection.

**Reasons To Reject:**

There are no particular grounds for rejection. However, it would have been interesting if this study had incorporated more non-proprietary models that could be easily reproduced for further investigation.

**Reproducibility:**

2: Would be hard pressed to reproduce the results. The contribution depends on data that are simply not available outside the author's institution or consortium; not enough details are provided.

**Reviewer Confidence:**

3: Pretty sure, but there's a chance I missed something. Although I have a good feel for this area in general, I did not carefully check the paper's details, e.g., the math, experimental design, or novelty.

**Typos Grammar Style And Presentation Improvements:**

I have two minor remarks regarding the presentation:

Figure 1: I understand that the purpose of this figure is to compare the trends of PaLM with previous models. However, I find the marker and line size to be slightly too large, making it difficult to discern the general trends of all the models.

Table 1: While I recognize that space constraints might have influenced its placement on the first page, I personally find that featuring something like the "average" subplot from Figure 1 would be more captivating for engaging a new reader.

---

> ### Author Rebuttal · Authors · 2023-08-28
>
> Thanks so much for your review!
>
> > However, it would have been interesting if this study had incorporated more non-proprietary models that could be easily reproduced for further investigation.
>
> We agree that it would be ideal to extend this study to non-proprietary models (although PaLM, while not completely open-source, has become accessible via the PaLM/Google Cloud API https://developers.generativeai.google/). Nevertheless, we point out that the largest non-proprietary models (in terms of # parameters) are BLOOM and OPT (~176B), which both fall in the range of model sizes already evaluated in the original inverse scaling paper (https://arxiv.org/abs/2306.09479), and therefore would not have been helpful in addressing the core research question of our paper (i.e., “Do models that are even larger than the largest models evaluated in the inverse scaling paper continue to show inverse scaling trends?”). In the future, we would love to extend our analysis to open-source models should larger models become available.
>
> > Have you verified whether the labels are evenly distributed in the prompts for each task, ensuring that the correct answer is not disproportionately positioned as the first or second option?
>
> We believe with the exception of Modus Tollens, the labels are evenly distributed. We also note that the tasks shown in Figure 5 are directly taken from the original inverse scaling paper (https://arxiv.org/abs/2306.09479) to maintain comparability with the previously reported results, and more justifications about the task/dataset design can be found in their respective descriptions in the said paper.
>
> > I have two minor remarks regarding the presentation:
>
> Thank you for the stylistic suggestions! We will try to incorporate them in the camera-ready version if the paper is accepted.

---

### Meta-Review · Area_Chair_t25w · 2023-09-18

**Recommendation:** 5

**Metareview:**

This paper presents a U-shaped scaling curve that identifies the "inverse scaling" behavior may not hold when the model continues to scale. Authors hypothesizes that medium-sized models are confused by task distractors, e.g. repetitive patterns, famous quotes, etc. In addition, authors find that one-shot prompts/chain-of-thought can mitigate these distractors. The findings in this paper are very interesting and experiments are well-executed.

---

### Decision · Program_Chairs · 2023-10-07

**Decision:**

Accept-Main

**Comment:**

This paper presents a U-shaped scaling curve that identifies the "inverse scaling" behavior may not hold when the model continues to scale. Authors hypothesizes that medium-sized models are confused by task distractors, e.g. repetitive patterns, famous quotes, etc. In addition, authors find that one-shot prompts/chain-of-thought can mitigate these distractors. The findings in this paper are very interesting and experiments are well-executed.